# Foldable interpenetrated metal-organic frameworks/carbon nanotubes thin film for lithium–sulfur batteries

Yiyin Mao[1], Gaoran Li[2], Yi Guo[1], Zhoupeng Li[2], Chengdu Liang[2], Xinsheng Peng[1] & Zhan Lin[2]

Lithium–sulfur batteries are promising technologies for powering flexible devices due to their high energy density, low cost and environmental friendliness, when the insulating nature, shuttle effect and volume expansion of sulfur electrodes are well addressed. Here, we report a strategy of using foldable interpenetrated metal-organic frameworks/carbon nanotubes thin film for binder-free advanced lithium–sulfur batteries through a facile confinement conversion. The carbon nanotubes interpenetrate through the metal-organic frameworks crystal and interweave the electrode into a stratified structure to provide both conductivity and structural integrity, while the highly porous metal-organic frameworks endow the electrode with strong sulfur confinement to achieve good cyclability. These hierarchical porous interpenetrated three-dimensional conductive networks with well confined $S_8$ lead to high sulfur loading and utilization, as well as high volumetric energy density.

[1] State Key Laboratory of Silicon Materials, School of Materials Science and Engineering, Zhejiang University, Hangzhou 310027, China. [2] Key Laboratory of Biomass Chemical Engineering of Ministry of Education, College of Chemical and Biochemical Engineering, Zhejiang University, Hangzhou 310027, China. Correspondence and requests for materials should be addressed to X.P. (email: pengxinsheng@zju.edu.cn) or to Z.L. (email: zhanlin@zju.edu.cn).

Flexible energy storage systems are undergoing blooming development due to the greatly raised demand of portable and wearable electronics[1–3]. Lithium–sulfur (Li–S) batteries, delivering high theoretical power density, low cost and environmental friendliness, possess great potential to meet the rapidly expanding market of flexible electronics[4–6]. Nevertheless, the poor electronic and ionic conductivities of the active material, the large volume change during charge–discharge, and the dissolution of intermediate polysulfides lead to poor lifespan of conventional Li–S batteries, restricting their large-scale production and commercial application[7,8]. Various strategies have been developed to improve the performance of Li–S batteries[9–12], yet the obtained volumetric energies are still low. Recently, metal-organic frameworks (MOFs) materials with abundant porosity, high specific surface area and strong chemical bondage with sulfur species revealed a great potential in tackling the polysulfide shuttle problem and enhancing electrochemical performance of Li–S batteries[13–19]. However, several challenges, for example, the electronic insulation (generally lower than $10^{-10}\,\mathrm{S\,cm^{-1}}$) and the brittleness of the MOFs materials, seriously constrain the kinetics of sulfur electrochemistry and mechanical stability of the electrode, leading to poor flexibility and cell performance which should be well addressed before the employment of MOFs for flexible Li–S batteries[20].

Herein, we present a MOFs/carbon nanotubes (CNT) thin film with unique hierarchical porous structure and interpenetrated three-dimensional conductive networks through a confinement conversion for flexible and even foldable Li–S batteries[21]. The abundant porosity with appropriate pore size, and the electrostatic attraction between negatively charged polysulfides and positively charged open metal sites in MOFs[19,22] provide strong confinement for sulfur species, while the interpenetrated CNTs contribute great conductivity inside and between MOF particles for sulfur electrochemistry, and weave the electrode into tough and pliable cloth to hold the large volume change during cycling, maintaining a great electrode structural integrity under bending and folding. The resultant Li–S batteries demonstrate good flexibility even foldability and excellent electrochemical performance, that is, great cycling stability for 500 cycles with 0.08% decay per cycle in coin cells and high volumetric energy density of $1,195\,\mathrm{mAh\,cm^{-3}}$ in soft package configuration. To the best of our knowledge, these results are amongst the best performances in MOFs-related Li–S batteries and flexible sulfur electrodes (Supplementary Tables 1–3).

## Results

**Synthesis and characterization of S@MOFs/CNT electrodes.** The foldable hierarchical porous MOFs/CNT composite thin films were prepared through a solid precursor assisted-confinement conversion process[21]. Metal hydroxide nanostrands (MHNs) with tiny diameter and highly positively charged surface[23] can easily assemble with negatively charged single-walled CNT by simply mixing[24]. MHNs/CNT composite thin films were obtained after filtering the mixed solution onto a porous substrate (Fig. 1, Supplementary Fig. 1a–d), which were subsequently immersed into organic ligand solution at room temperature for 1 h to achieve flexible MOFs/CNT composite thin films.

To investigate the MOFs pore size effect on Li–S performance, HKUST-1 (($Cu_3(BTC)_2$, cavity of 1.1 nm with 0.9 nm entrance and active copper sites; BTC: 1,3,5-benzenetricarboxlic acid)[25], MOF-5 (($Zn_4O(BDC)_3$, cavity of 1.5 nm with 0.8 nm entrance; BDC: benzene-1,4-dicarboxylic acid)[26], and ZIF-8 (($Zn(mim)_2$, cavity of 1.16 nm with 0.34 nm entrance; mim:

2-Methylimidazole)[27] based CNT composite thin films were prepared (Fig. 1, Supplementary Fig. 2). The HKUST-1/CNT composite thin film with HKUST-1 to CNT weight ratio of 3:2 shows typical stratified structures with good self-standing property and foldability (Fig. 2a–c). The HKUST-1 crystal size ranges from 500 nm to 2 $\mu$m, the CNT penetrate through MOFs particles (Fig. 2d, Supplementary Fig. 3) and weave them into a tough thin film. The thickness of the composite film is 13.5 $\mu$m (Fig. 2c), which is much thicker than the copper hydroxide nanostrands (CHNs)/CNT precursor thin film (8.25 $\mu$m, Supplementary Fig. 1). The three-dimensional CNT networks penetrate through and connect the HKUST-1 crystals together, thus guaranteeing the great conductivity and the foldable properties of the corresponding electrodes (Fig. 2, Supplementary Movie 1). Similar hierarchical porous structures are also observed in ZIF-8/CNT (Supplementary Fig. 2a,b) and MOF-5/CNT thin films (Supplementary Fig. 2c,d) with MOFs to CNT ratio of 3:2. These foldable layered hierarchical porous MOFs/CNT thin films with good conductivity are promising for binder-free Li–S batteries.

As well documented, $S_8$ with molecular size of 0.68 nm is the major existence of sulfur at temperature no higher than 140 °C (ref. 28), which delivers the highest power density among sulfur species[7–12]. Here $S_8$ was loaded by infiltering sulfur $CS_2$ solution into activated HKUST-1/CNT, MOFs-5/CNT and ZIF-8/CNT thin films, drying under vacuum at 60 °C, and then keeping at 140 °C for 8 h (see detailed information in 'Methods' section). The corresponding electrodes are denoted as S@KHUST-1/CNT, S@MOF-5/CNT and S@ZIF-8/CNT, respectively. The structures of the MOFs/CNT thin films and the MOFs crystal are well remained after $S_8$ loading without obvious sulfur particles (Fig. 2e,f, Supplementary Fig. 2e,f). As demonstrated in Fig. 1, $S_8$ can easily access into the cavities of MOF-5 and HKUST-1 through their big entrances, while it is hard to reach into cavities of ZIF-8 due to its relatively small entrance (0.34 nm). These results were confirmed by the energy dispersive X-ray spectroscopy (EDX, Supplementary Fig. 4). Elemental sulfur was well found in HKUST-1 (Supplementary Fig. 4a,b) and MOF-5 (Supplementary Fig. 4c,d) after washing by $CS_2$, while much less sulfur species were retained for ZIF-8 crystals (Supplementary Fig. 4e,f). The variation of peak intensities (Supplementary Fig. 4g) and element contents (Supplementary Table 4) before and after $CS_2$ washing also strongly support this result. The typical sulfur mass content in S@MOF/CNT electrode for coin cells cycling is ~40 wt% confirmed by the thermo graimetric analyzer (TGA) analysis (Supplementary Fig. 5a). The corresponding XRD results (Supplementary Fig. 2e,f) reveal no obvious crystalline sulfur was formed after sulfur loading due to good pore occupation by sulfur in MOFs, which is also confirmed by $N_2$ adsorption/desorption isotherm analysis (Supplementary Fig. 5b).

**Electrochemical characterization.** The electrochemical performances of the electrodes based on HKUST-1 and/or CNT in coin cells configuration with a sulfur loading of $1\,\mathrm{mg\,cm^{-2}}$ (~40 wt%) are shown in Supplementary Fig. 6. Prepared by conventional slurry-casting process, the cycling capacity of the S@CNT electrode exhibits more rapid decrease especially in the first few cycles than that of the S@HKUST-1 + CNT electrode, in which the HKUST-1 particles and CNT are simply mixed together in weight ratio of 3:2. Meanwhile, the S@HKUST-1 electrode exhibits more stable cycling behaviour but lower capacity than the S@HKUST-1 + CNT electrode due to the poor electronic conductivity and low sulfur utilization. In contrast, the as-prepared self-standing S@HKUST-1/CNT electrode delivers obviously higher capacity and cycling stability than all the

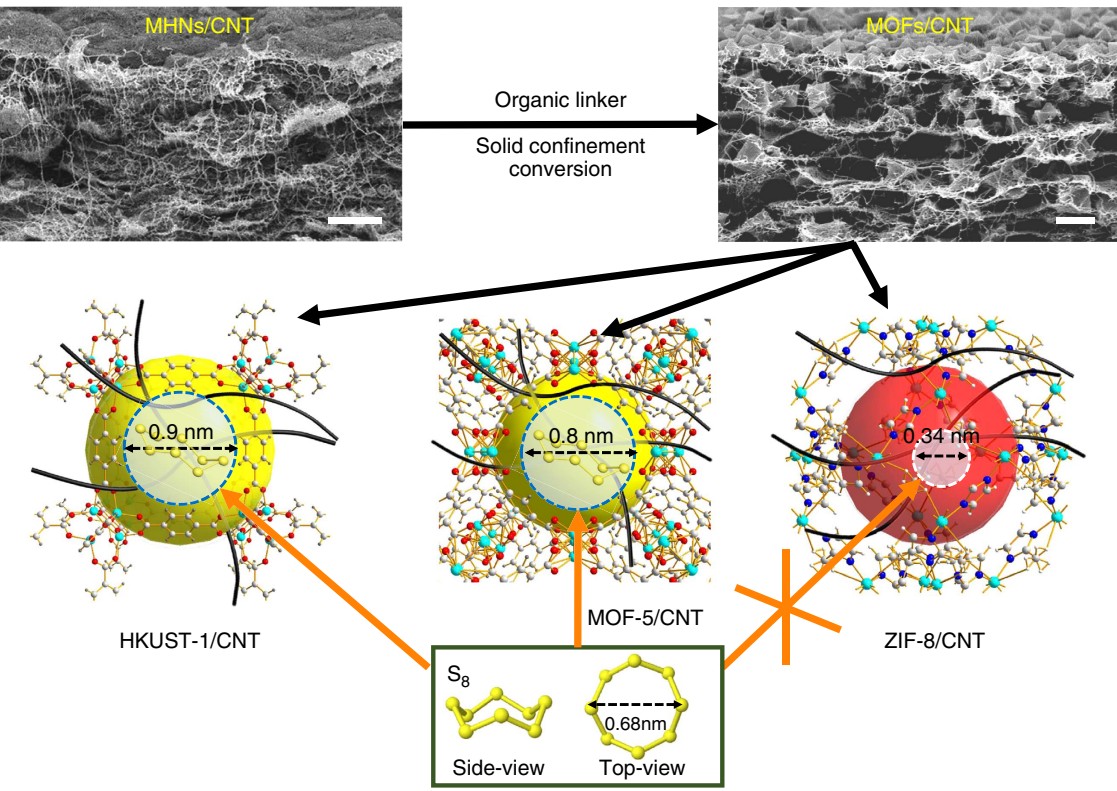

**Figure 1 | Synthesis of S$_8$ loaded MOFs/CNT composite thin films.** The scale bars in the s.e.m. images are 1 μm. MHNs, CNT, MOFs represent the metal hydroxide nanostrands, carbon nanotubes and metal-organic frameworks, respectively.

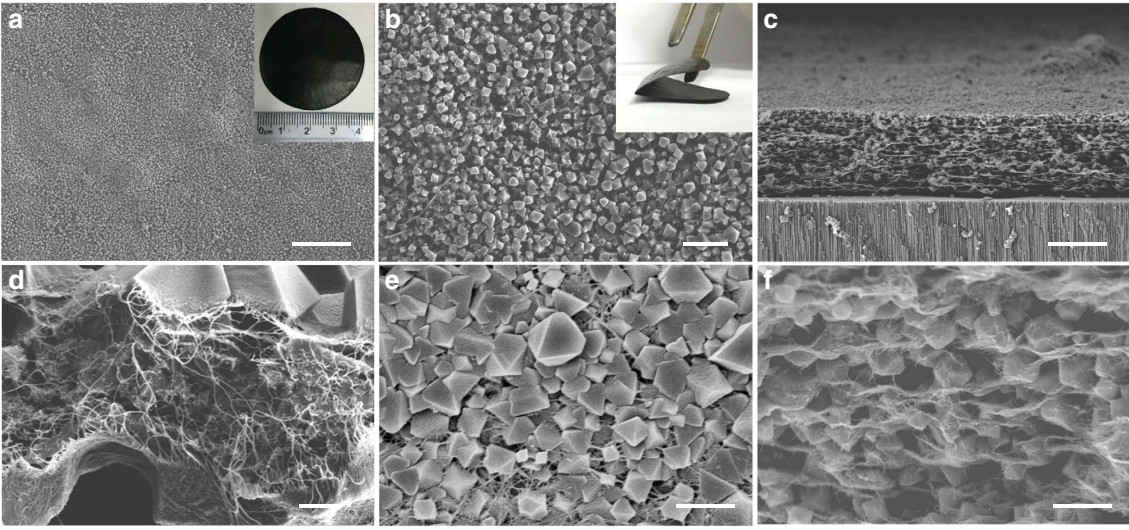

**Figure 2 | Morphology of porous HKUST-1/CNT composite thin film.** (**a**,**b**) surface and (**c**,**d**) cross-section s.e.m. images before sulfur loading. (**e**) Surface and (**f**) cross-section SEM images after sulfur loading. The insets in **a** and **b** are corresponding photographs of HKUST-1/CNT thin film in diameter of 4 cm with HKUST-1 to CNT of 3:2. Scale bars, 20 μm (**a**), 3 μm (**b**), 10 μm (**c**), 500 nm (**d**), 1 μm (**e**) and 2 μm (**f**).

casting-processed electrodes, indicating the strong superiority of its advanced electrode structure in performing the sulfur electrochemistry (Supplementary Fig. 6a). The electrochemical impedance spectroscopy (EIS) results (Supplementary Fig. 6b) well consistent with the cycling performances that the self-standing S@HKUST-1/CNT electrode exhibits smaller impedance than those prepared by slurry-casting process. The galvanostatic charge–discharge cycling performances of the electrodes with different HKUST-1 to CNT weight ratios (Supplementary Fig. 6c) reveal that the cycling stability is improved with increase in the content of HKUST-1 owing to enhanced confinement to sulfur species, while the capacity is significantly limited due to poor electrode conductivity resulting from the insufficiency of conductive CNT network. These are also confirmed by corresponding scanning electron microscopy (SEM) images (Supplementary Fig. 7). The optimal weight ratio of HKUST-1 to

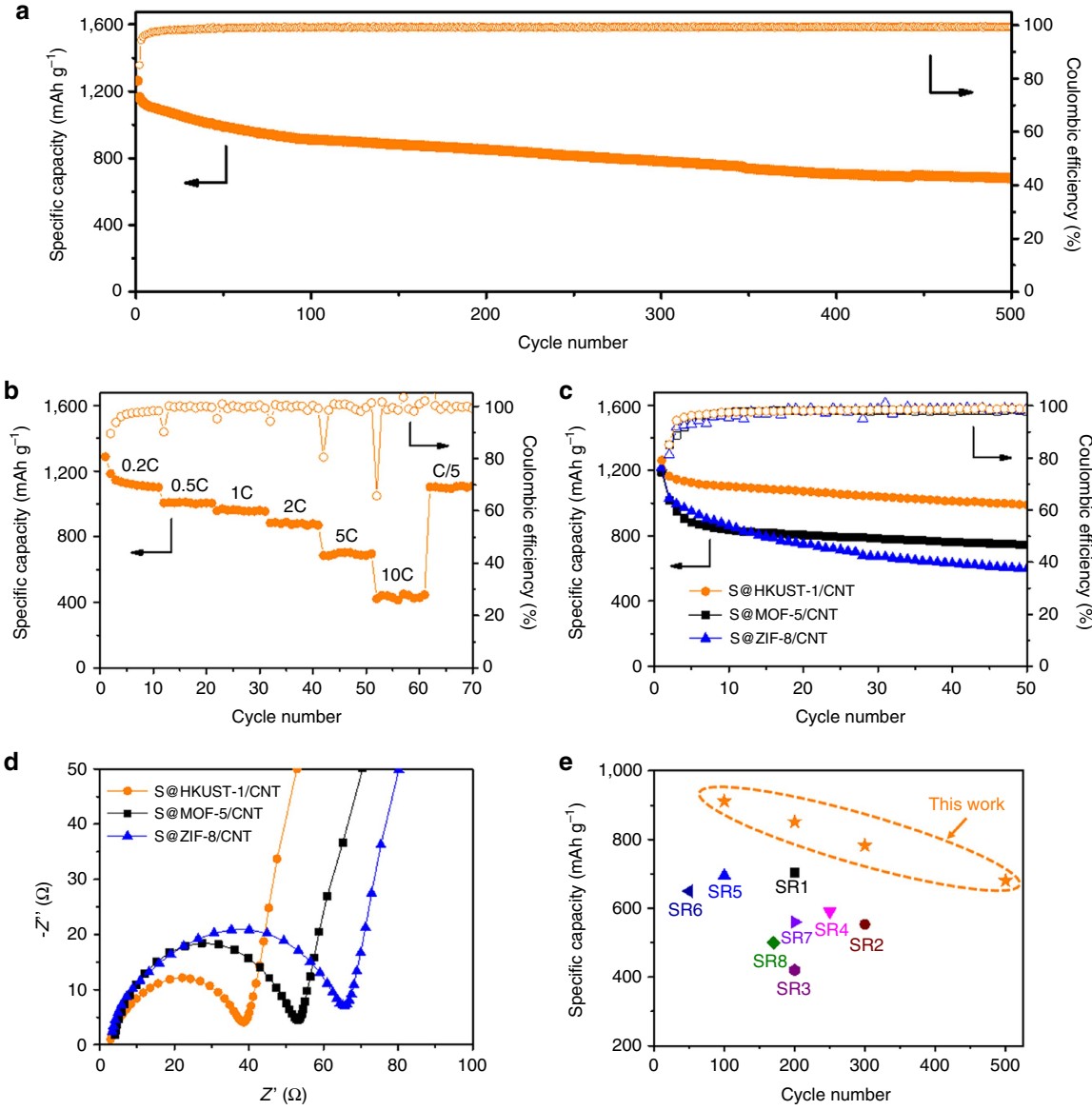

**Figure 3 | The electrochemical performance of S@MOF/CNT electrodes. (a)** The cycling and (**b**) rate performances of S@HKUST-1/CNT electrode. (**c**) The cycling performances and (**d**) Nyquist plots of S@HKUST-1/CNT, S@MOF-5/CNT and S@ZIF-8/CNT electrodes, respectively. (**e**) The capacity comparison between S@HKUST-1/CNT electrode and the MOFs-based sulfur electrodes in representative publications (see details in Supplementary Table 1, Supplementary References SR1–8). The sulfur loading is $1 \, mg \, cm^{-2}$ and MOFs to CNT ratio is 3:2.

CNT is 3:2, which exhibits the best cycling performance among the compared electrodes. The MOFs to CNT weight ratio of all the after mentioned MOFs/CNT electrodes is 3:2. Supplementary Fig. 8a shows the typical two-platform discharge profile of elementary $S_8$ for S@HKUST-1/CNT electrode, corresponding to the electrochemical reduction of $S_8$ to soluble long-chain polysulfides ($Li_2S_x$, $x = 4–8$) and subsequently to insoluble $Li_2S_2/Li_2S$, respectively[9–12]. The cyclic voltammetry curves (Supplementary Fig. 8b) show excellent consistency with the charge–discharge profile.

The self-standing S@HKUST-1/CNT electrode with $1 \, mg \, cm^{-2}$ sulfur loading and HKUST-1 to CNT ratio of 3:2 delivers a high initial capacity of $1,263 \, mAh \, g^{-1}$ at 0.2 C ($1 \, C = 1,675 \, mA \, g^{-1}$, Fig. 3a), which is as good as those reported before[29–31]. Moreover, excellent cyclability over 500 cycles with a small fading rate of 0.08% per cycle at 0.2 C and high coulumbic efficiency above 99% are also achieved.

Excellent rate capability is also achieved for S@HKUST-1/CNT electrode with a highly reversibly capacity of $880 \, mAh \, g^{-1}$ at 2 C and $449 \, mAh \, g^{-1}$ at an even high rate of 10 C. The capacity recovers to $1,102 \, mAh \, g^{-1}$ when the current rate goes back to 0.2 C (Fig. 3b), indicating the good electrode integrity and fast reaction kinetics of the electrode. After 200 times deep cycling, the stratified structure, crystal phase and morphology of HKUST-1 crystals are well remained as confirmed by SEM and XRD test (Supplementary Fig. 9). The electrochemical performances of the S@MOF-5/CNT and S@ZIF-8/CNT electrodes with MOF to CNT weight ratio of 3:2, and sulfur loading of $1 \, mg \, cm^{-2}$ were also investigated (Fig. 3c,d, Supplementary Fig. 8c,d). The S@MOF-5/CNT electrode exhibits similar stable cycling behaviour as S@HKUST-1/CNT electrode after a severe capacity loss in the first few cycles, while the S@ZIF-8/CNT electrode shows much faster capacity fading during cycling. The EIS results (Fig. 3d) with an impedance

order of S@ZIF-8/CNT > S@MOF-5/CNT > S@HKUST-1/CNT electrode support the cycling performance comparison. This is mainly attributed to the size sieving effect for different MOFs materials.

As mentioned above, sulfur is loaded into the corresponding MOFs thin films in the form of $S_8$ ring with 0.68 nm diameter[28], which is hardly introduced into the ZIF-8 cavity with 0.34 nm entrance[27], and mainly disperse on the surface of ZIF-8 particles. This leads to low sulfur utilization, poor reaction kinetics and fast capacity fading. In contrast, HKUST-1 and MOF-5 possess relatively larger entrance size than $S_8$ molecules, providing easy access for them into pores, which is beneficial for uniform sulfur incorporation and electrochemical reaction. Moreover, the success in sulfur and polysulfides confinement active copper sites improves electrode cyclability by inhibiting active sulfur loss[32,33]. Compared to HKUST-1, MOF-5 has slightly smaller entrance size, which results in relatively less $S_8$ loaded inside the pores and more on its outside surface, leading to faster capacity decay during the first few cycles (Fig. 3c). The entrance size effect was confirmed by the EDX characterization (Supplementary Fig. 4). After $CS_2$ washing, the electrodes possess the sulfur retainment in order of HKUST-1 > MOF-5 ≫ ZIF-8, which strongly reveals the importance of appropriate pore size in retaining active sulfur. Considering the small pore size of ZIF-8, the S@ZIF-8/CNT electrode might be good for active sulfur in the form of small molecules, namely $S_x$ ($x = 2$–4)[34]. In addition, another effect should also be taken into consideration that the open copper metal sites in HKUST-1 could interact with polysulphides, while there are no open metal sites in MOF-5 and ZIF-8 (refs 13,19,22). This also contributed to the superior cell performance of S@HKUST-1/CNT electrode. It should be noted that the success in electrode fabrication and cell cycling for the flexible electrodes with different MOF materials demonstrates the universality and practicality of this strategy.

## Discussion

In general, excellent electrochemical performance was achieved for the as-prepared S@HKUST-1/CNT electrode through this strategy. To the best of our knowledge, the cycling and rate performance of S@ HKUST-1/CNT electrode is the best performance among MOFs-based sulfur electrodes[13–19] (Fig. 3e, Supplementary Table 1) and competitive to those of representative high-performance carbon-based sulfur electrodes[5,9–12,34–55] (Supplementary Table 2). The excellent performance is attributed to the following reasons. Firstly, the polysulfides and active sulfur are well confined in HKUST-1 due to its abundant porosity with appropriate pore size, and the electrostatic attraction between negatively charged polysulfides and positively charged open copper sites after dehydration[13]. Secondly, the CNTs weave the electrode into pliable and tough cloth to hold large volume change during cycling and keep electrode structural integrity under bending and folding. Thirdly, MOFs are also endowed with favourable tolerance to volume change to some extent[56]. Moreover, the CNTs interpenetrated HKUST-1 crystal in molecular scale guarantee the easy access of electron to the trapped sulfur species, while the numerous conductive linkages between MOF particles provide good electron transport within the electrode. Finally, plenty of unique hierarchical nanoporous and macroporous structures provide great electrolyte wettability to ensure $Li^+$ supply for sulfur electrochemistry.

To verify the practicality of the as-prepared electrode for flexible Li–S batteries, a rectangle shape soft package Li–S cell based on S@HKUST-1/CNT electrode was prepared. The obtained soft package Li–S cell is capable to lighten up an LED in various folding angles, for example, 0°, 90°, 180° and back to 0° (Fig. 4a, Supplementary Movie 2). Figure 4b shows the stable cyclability of the folded cell in different angles, with slight capacity decay along with the increase of folded extent. The areal capacity rebounds to ~3.5 mAh cm$^{-2}$ when the cell returns flat, indicating the excellent reversibility of cell performance against cell deformation extent. Meanwhile, the band shape cell was also prepared and used to light up a LED when bent to different extents, for example, flat, slightly bent, circled and back to flat (Supplementary Fig. 10). The EIS results demonstrate that the cell impedance gradually increased with the folding degree, as well as a certain increase after 50 cycles (Supplementary Fig. 11). When the cell recovers to flat state, the impedance decreases back. The long-term galvanostatic cycling test up to 200 cycles with a sulfur loading of 4.57 mg cm$^{-2}$ shows that the areal capacity of the soft-packed cell soon reaches 3.5 mAh cm$^{-2}$ after the activation in the first few cycles, higher than that of the commercialized Li-ion battery (around 3.0 mAh cm$^{-2}$)[43], and remains stable during the cycles afterwards (Fig. 4c). These results indicate the excellent flexibility and stable electrochemistry of the soft package Li–S batteries based on the S@HKUST-1@CNT electrodes, revealing the great potential for its practical application in flexible electronics.

Higher areal sulfur loadings on S@HKUST-1/CNT electrodes were also prepared by raising the electrode thickness (Supplementary Fig. 12), while keeping the sulfur content of about 70 wt% for higher energy density of soft package Li–S cells. The fabricated Li–S cells achieved a high capacity of ~7.45 mAh cm$^{-2}$ and favourable cyclability when sulfur loading reached as high as 11.33 mg cm$^{-2}$ (Fig. 4d, Supplementary Fig. 13). Owing to advanced electrode structure and controlled electrode thickness, this strategy offers an outstanding cell performance, which presents the highest volumetric capacity among the recently reported flexible sulfur cathodes (Fig. 4d, Supplementary Table 3)[38–41], revealing a great practical potential for next-generation high power density flexible electronics. However, the cycling performance of sulfur electrodes with high volumetric energy densities can be further improved by finding MOFs with proper entrance pore size, big pore volume and plenty of open metal sites, which not only let solvated Li ions transfer in and out but also allow sulfur species to enter and remain confined in the pore for good inhibition of the polysulfide shuttle.

To sum up, we developed a facile method to prepare CNTs interpenetrated MOFs thin film with unique hierarchical porous structures for binder-free, flexible, and even foldable Li–S batteries with high volumetric energy density and long lifespan. Our design fulfils the most rigorous requirements of flexible Li–S batteries, that is, maintaining electrical connectivity of confined active sulfur, tolerating large volume change during lithiation/delithiation and endowing the electrode with great flexibility and integrity through CNTs interpenetrated MOFs. As a result, it is possible to design MOFs-based flexible energy storage systems with high volumetric energy density for possible applications in portable and flexible electronics. This method opens up the possibility to design other high energy density rechargeable lithium batteries, such as lithium-ion, sodium-ion and lithium-air batteries, with good flexibility and long lifespan in the near future.

## Methods

**Preparation of MHNs.** Copper hydroxide nanostrands were synthesized by quickly mixing equal volume 4 mM copper nitrate aqueous solution with 1.6 mM aminoethanol aqueous solution at room temperature and aged for 2 days. Zinc hydroxide nanostrands were synthesized by quickly mixing 4 mM zinc nitrate aqueous solution with 2.0 mM aminoethanol aqueous solution at room temperature and aged for 30 min.

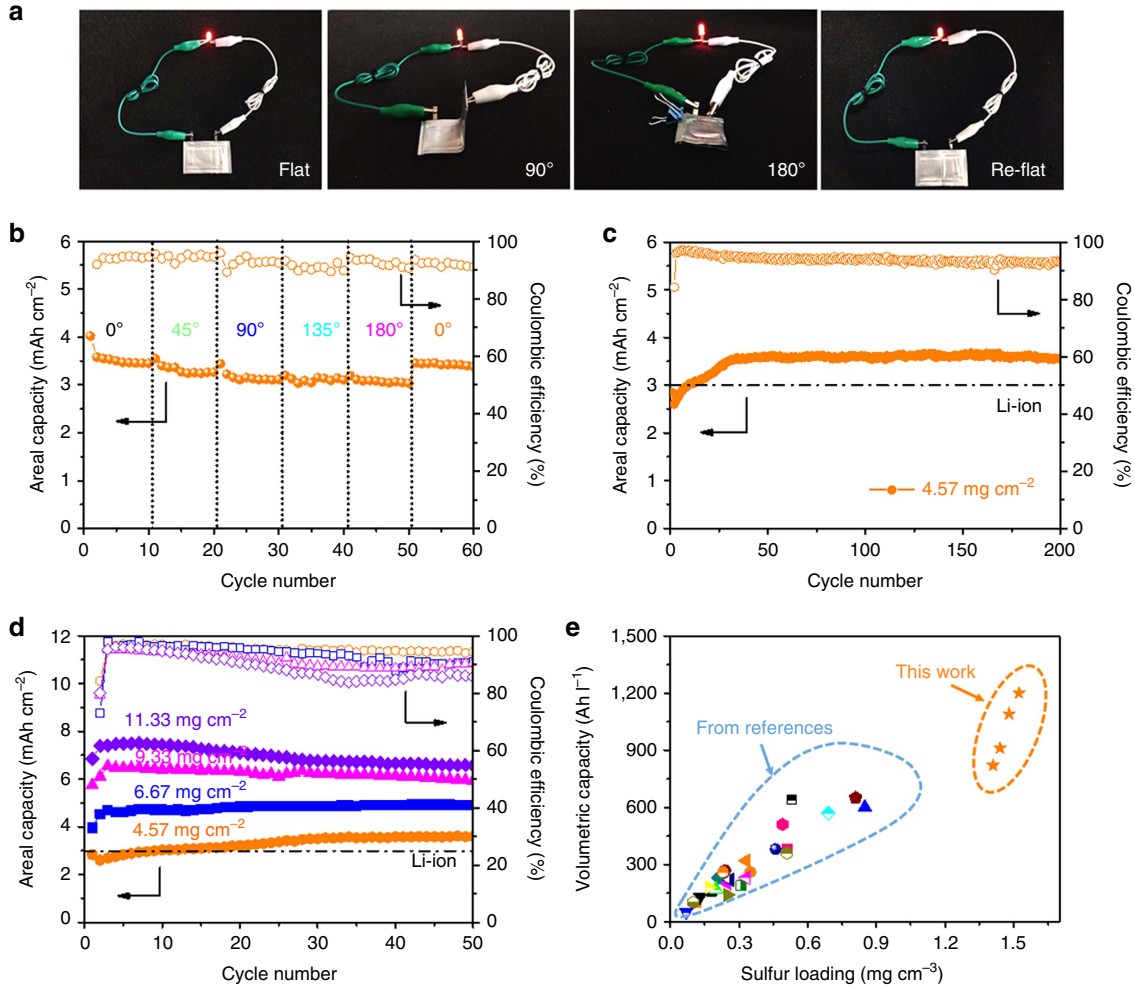

**Figure 4 | The electrochemical performance of soft package Li–S cells.** (**a**) The photograph of soft package Li–S cell lighting up an LED under different folding angles. (**b**) The cycling performance of soft package Li–S cell under different angles. (**c**) The long-term galvanostatic cycling up to 200 cycles with a sulfur loading of $4.57\,mg\,cm^{-2}$. (**d**) The cycling performance of soft package Li -S cell with different sulfur loadings. (**e**) The volumetric capacity comparison between S@HKUST-1/CNT electrode and the self-standing and /or flexible sulfur electrodes in representative publications (supplementary table 3 and supplementary references SR20, SR40 -64).

**Preparation of negatively charged single-walled CNTs.** Negatively charged CNTs were prepared by partial oxidation using nitric acid. A certain amount of single-walled CNT was heated in 6 M nitric acid at 80 °C for 24 h. The solution was subsequently washed with adequate deionized water, and the precipitates were collected by centrifugation. The washing and centrifuge were repeated several times until the solution reached to pH 6.5. The resultant materials were dispersed in water to give a concentration of $0.9\,mg\,ml^{-1}$.

**Preparation of HKUST-1/CNT composite thin film.** On the highly positively charged surface of MHNs, the negatively charged CNTs easily connect with the MHNs by simply mixing. After filtering the mixed solution onto a porous substrate, MHNs/CNT composite thin films were obtained. Then flexible MOFs/CNT composite thin films were prepared by immersing MHNs/CNT composite thin films into certain organic ligand solution at room temperature for 1 h. By controlling the weight ratio of MHNs to CNTs, MOFs/CNT thin films with different weight ratios of MOFs to CNT were obtained. The diameter of the films was determined by the funnel size. For example, after filtering the solution mixed by 30 ml CHNs and 1.5 ml CNTs ($0.9\,mg\,ml^{-1}$) on a porous anodic alumina oxide (AAO) membrane with effective diameter of 2 cm and pore size of 200 nm, a CHNs/CNT thin film with about 8.25 μm was obtained. This CHNs/CNT composite thin film was reacted with 10 ml, 20 mM $H_3BTC$ water/ethanol solution with volume ratio of 1:1 at room temperature, and converted to HKUST-1/CNTs composite thin film with thickness of about 13.5 μm. The thickness of the final film was linearly increased with the thickness of the CHNs/CNT precursors when keeping other conditions the same. After dissembling HKUST-1/CNT in diluted 10 mM HCl solution, CNT was collected by filtering the solution through an AAO substrate, the concentration of Cu in the filtrate was determined by inductively coupled plasma mass spectrometer (ICP-MS, Agilent

G3271A). The final weight ratio of HKUST-1 ($Cu_3(BTC)_2$) to CNT in HKUST-1/CNT prepared from 30 ml CHNs solution and 1.5 ml $0.9\,mg\,ml^{-1}$ CNT dispersion is 3:2. Similarly, flexible HKUST-1/CNT composite thin films with different weight ratios of HKUST-1 to CNT were prepared to investigate the optimal conditions by controlling the amount of CHNs and CNT. The thickness of the HKUST-1/CNT thin films linearly depends on the volumes of the CHN solution and CNT dispersion, when the diameter of the funnel is the same. Based on this strategy, thicker electrodes can be prepared easily by either raising raw material amounts and reaction times, or simply stacking thin films. However, with increase in thickness of the electrode, it would result in relatively poor active material utilization due to longer electronic and ionic transfer in the vertical direction of the electrode.

**Preparation of ZIF-8/CNT and MOF-5/CNT thin films.** ZIF-8/CNT thin film with weight ratio of ZIF-8 to CNT close to 3:2 was prepared by filtering the mixed 30 ml ZHNs and 2.0 ml CNTs $0.9\,mg\,ml^{-1}$ solution on AAO substrate with effective area of 2 cm, and reacting in 10 ml, 25 mM Hmim ethanol/water solution with volume ratio of 1:4 at room temperature for 24 h. While MOF-5/CNTcomposite thin film with weight ratio of MOF-5 to CNT close to 3:2 was prepared by filtering the mixed 30 ml ZHNs and 1.8 ml CNTs $0.9\,mg\,ml^{-1}$ solution on AAO substrates, and reacting in 10 ml, 20 mM $H_2BDC$ dimethylformamide solution at 120 °C for 12 h.

**Preparation of S@MOFs/CNT electrodes.** Typically, sulfur $CS_2$ solution ($5\,mg\,ml^{-1}$) was first prepared by dissolving sublimed sulfur in $CS_2$ solvent under stirring at room temperature. The obtained solution was dropped into the MOFs/CNT substrates and slowly dried at 40 °C. The sulfur contained film was further dried under vacuum at 60 °C for 8 h and subsequently annealed at 140 °C

for 8 h under Ar atmosphere and cooled to room temperature to obtain the S@HKUST-1/CNT, S@MOF-5/CNT and S@ZIF-8/CNT electrodes, respectively. The sulfur loading amount was controlled by the dropping volume of sulfur solution, and obtained by weighting MOFs/CNT thin films before and after loading sulfur. The areal sulfur loading was controlled by the thickness of the MOFs/CNT films and the dropped volume of the sulfur $CS_2$ solution. Typically, the areal loading amount of sulfur for coin cells was about $1 \, mg \, cm^{-2}$ for a 22.5 μm thick film, which is common loading used for cycling performance[29–31,45,57–62]. The sulfur content in electrodes for coin cells is ~40 wt%, which is a common value for MOFs-based sulfur electrodes[16,18,19]. However, the sulfur contents as high as ~70 wt% (with the content of CNT ~12%), were also achieved for electrodes in soft package cells, as shown in Supplementary Table 5. For comparison, sulfur electrodes containing HKUST-1 and/or CNT were also prepared through the conventional slurry-casting process. Slurry containing 40 wt% sulfur and 10 wt% PVDF with 30 wt% HKUST-1 + 20 wt% Super P (S@HKUST-1), or 50 wt% CNT (S@CNT), or 30 wt% HKUST-1 + 20 wt% CNT (S@HKUST-1 + CNT) was prepared in N-methyl-2-pyrrolidone (NMP) solvent, casted on the Al foil and vacuum dried at 60 °C for 12 h. In electrode with thickness of 22.5 μm and sulfur loading of $1 \, mg \, cm^{-2}$, the micropores are not fully filled when the sulfur content is about 40 wt%. To achieve higher sulfur content and electrode capacity in Li–S soft package configuration, we only raised the electrode thickness from 22.5 to 30 μm when sulfur loading increased from 1 to $4.57 \, mg \, cm^{-2}$, thus the sulfur content was increased from 40 to 70 wt%. However, in attempt for even higher sulfur loading and areal capacity, we increased the electrode thickness from 30.4 to 44.6, 64.7, and 80.0 μm along with the increase of areal sulfur loading from 4.57 to 6.67, 9.33, and $11.33 \, mg \, cm^{-2}$, but maintained the sulfur content to ~70 wt% for decent conductivity and sulfur confinement. The resultant volumetric sulfur loading was 1.52, 1.48, 1.43 and $1.41 \, g \, cm^{-3}$, respectively. Summarized information is shown in Supplementary Table 5.

**Characterization.** The XRD of as-prepared products were characterized at room temperature using an X'Pert PRO (PANalytical, Netherlands) instrument with Cu Kα radiation. The morphologies and structures were characterized by using SEM (Hitachi S-4800) with EDX. SEM observation was conducted after coating a thin platinum layer by using a Hitachi e-1030 ion sputter at the pressure of 10 Pa and the current density of 10 mA.

**Electrochemical measurements.** The electrochemical performances of the S@MOFs/CNT electrodes were valued in coin cells and soft package cells. The CR2025 coin cells were assembled in an Ar-filled glove box using metallic lithium wafer as counter electrode and Celgard 2400 membrane as separator. The electrode and separator did not stick together, and there was no problem for peeling-off between the electrode and the separator in the cells. As for aluminium laminated soft-packed cells, the S@HKUST-1@CNT electrode was cut into 2 cm × 3 cm rectangle slice for rectangle shape cells and 1.5 cm × 4.0 cm for band shape cells. The corresponding shaped Celgard membrane and lithium foil were used as separator and counter electrode, respectively. The electrolyte contains 1 M lithium bis(trifluoromethane sulfonyl) imide (LiTFSI) in a binary solvent of 1,3-dioxolane and dimethoxyethane (1:1 in volume) with 1 wt% lithium nitrate as the additive. When bending or folding occurs to the cells with limited electrolyte, the electrolyte held by electrode and separator tends to be squeezed aside, leading to relatively poor electrolyte wetness and increased capacity loss. As comparison, when cells are flooded by the electrolyte, the effect mentioned-above is strongly minimized. However, the excessive electrolyte aggravates the polysulfide shuttle effect with fast capacity decay. As a result, the electrolyte to sulfur (E/S) ratio was controlled at 50 μl electrolyte per mg sulfur for coin cells and 0.3 ml per cell for soft package cells. The cycling performance of the fabricated cells were tested by galvanostatic charge/discharge at room temperature using LAND battery cycler (China) within the voltage window of 1.7–2.8 V versus $Li/Li^+$ for coin cells and 1.5–3.0 V versus $Li/Li^+$ for soft-package cells to overcome relatively large overpotential. The rate capacity was also tested with current rates varied from 0.2 C to 10 C. Current density and specific capacity were calculated based on the mass of sulfur active material. The volumetric capacities are based on the volume of sulfur electrodes. Cyclic voltammetry study of the electrode was recorded by a CHI660E electro-chemical work station (Chinstruments, China) in the voltage range of 1.7–2.8 V versus $Li/Li^+$ at a scan rate of $0.1 \, mV \, s^{-1}$. EIS of the electrode was recorded by a CHI660E electrochemical work station with amplitude of 5 mV in the frequency range of 0.01 Hz–100 kHz.

**Data availability.** All relevant data supporting the findings of this study are available from the corresponding authors on request.

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

## Acknowledgements

This work is supported by the National Basic Research Program of China 973 Program (2015CB655302), National Key Research and Development Program (2016YFA0200204), the Natural Science Foundation for Outstanding Young Scientist of Zhejiang Province, China (LR14E020001) and the National Natural Science Foundations of China (NSFC 21671171). Z. Lin thanks the funding support from Chinese government under the 'Thousand Youth Talents Program' and from Zhejiang Province Science Fund for Distinguished Young Scholars (Project LR16B060001).

## Author contributions

Y.M. and G.L. contributed equally to this work. Original idea was conceived by X.P., Y.M. and G.L.; experiments and data analysis were performed by Y.M. and G.L.; the manuscript was drafted by Y.M., G.L., Z.L. and X.P. All authors have given approval to the final version of the manuscript.

## Additional information

**Competing financial interests:** The authors declare no competing financial interests.

