## [Peer Review File · Nature Communications]

Reviewers' comments:

Reviewer #1 (Remarks to the Author):

This manuscript communicates the making of binder free cathode made from MOFs interpenetrated by CNT. CNT provides the conductivity and also acts as binder, meanwhile they also act as network blocking the lithium sulfides from leaving the MOF cages. The electrode is made flexible with 90 folding causing no significant detrimental effects. The results are quite impressive in terms of capacity and also rate performance for HKUST-1/CNT system. The paper can be accepted for publication with the following issues addressed:

- 1) The pore size differences in MOFs were found to affect the loading of S8 with larger pores giving better loading as expected. The large pores would have higher chance of lithium sulfides to leak into the electrolyte. Do the authors find this?
- 2) For the CNT utilized, do authors find if there are any sulfur inside the tubes? If yes, how much of the contribution is from CNT and how much from MOF?
- 3) The cycling performance is ok but not fabulous. Would there be any way to improve it?

Reviewer #2 (Remarks to the Author):

In this paper, the authors disclosed a new strategy to interpenetrate the MOFs with CNTs as a foldable and bendable film for the cathode of Li-S batteries. By virtue of the electrostatic interaction between MOFs and polysulfides, the porous structure of MOFs, and the integrity of interweaved MOF/CNT film, the performances of Li-S batteries could be effectively enhanced. In addition, the film is binder-free, which is beneficial for the energy density of Li-S batteries. The experimental results are interesting and impressive. The novelties include: (i) this study made a comparison of three MOFs with different pore window and claimed HKUST-1 has suitable pore size to store S8 molecule; and (ii) the Li-S batteries showed the best reported performance in MOF based sulfur electrode. Based on the novelties and current quality of your manuscript, I think that this manuscript can be published on Nature Communications after some suggestions and comments are addressed:

- Technical questions

(1) Does CNTs interpenetrate through MOFs? From your SEM image (Fig.2d), it looks like the CNT only weaves MOF crystals and doesn't penetrate through the MOFs. If so, kindly provide cross-section TEM images showing the situation. All your discussions are based on the structure, you could better make it reliable and dependable.

(2) As described, elementary sulfur exists in the cavity of MOFs, especially in the cases of HKUST-1 and MOF-5, and CNTs weaves the MOF crystal (500 nm ~ 2 μm), how come the S8 inside the MOFs gets the electrons and Li⁺ ions to go through electrochemical reactions? Can the introduction of CNTs dramatically improve the conductivity of MOF matrix? In my opinion, MOFs are extremely insulative (10⁻¹⁰ S/cm), the distance of electron tunneling in MOFs is also very short (at the nanometer long). So the only reactive sites at the interface between CNTs and MOF crystals could be useful for the electrochemical reactions.

(3) What is the amount of CNTs in the sample of S@HKUST-1/CNT with 70% sulfur loading for batteries? Both sulfur and MOF are highly non-conductive, without enough CNTs as conductive networks, please explain how come the electrode can deliver a high capacity over 500 mAh/g (Table S4)?

(4) For element mapping images shown in Figure S3, the corresponding EDX analyses should be

accompanied as well, because the elemental relative content is very sensitive to the time to collect the characteristic X-ray signal. Technically and precisely, only elemental mapping images cannot demonstrate that sulfur is less in S@ZIF-8/CNTs after CS₂ washing. Besides, a question for the method of CS₂ washing is raised up, the sulfur incorporation into the cavities of MOFs was achieved by CS₂ solution. To remove the sulfur dispersed on the surface of S@ZIF-8/CNTs, the same method is also used. How can you ensure that the sulfur inside the cavities of composites was not washed? I think the sulfur can go in and also can go out providing using the same method.

(5) In Figure S5(c), there is a discrepancy. The cycling stability of S@HKUST-1/CNT (3:0.4) is worse than that of S@HKUST-1/CNT (3:1.2). However, the relative amount of HKUST-1 of the former is higher than that of the latter, meaning S@HKUST-1/CNT (3:0.4) should have better capabilities in confining polysulfides. As a consequence, S@HKUST-1/CNT (3:0.4) should show the better cycling stability than that of S@HKUST-1/CNT (3:1.2), which is discrepant from the results shown in Figure S5(c). Please double check.

(6) You defined 1C = 1675 mAh/g, the unit should be "mA/g" not "mAh/g". please correct it.

- Grammatical and spelling mistakes

There are a lot of sloppy grammatical and spelling mistakes everywhere in this manuscript. Please pay particular attention to your English grammar, spelling, and sentence structure. I listed out some of them for your consideration.

(7) In the line of 14 in the abstract, "candidate" should be "candidates" or "a promising candidate" because this is a countable noun.

(8) In the line of 24, "A, B and C" should be "A, B, and C". In the line of 27, "A, and B" should be "A and B", It don't need a comma between A and B.

(9) In the line of 27, "Flexible energy storage system" should be "A flexible energy storage system" or "Flexible energy storage systems". Because "system" is a countable nouns

(10) In the line of 46, "flexible even foldable" should change into "flexible and even foldable"

(11) In the line of 62, there is a typo error, "tinny diameter" should be "tiny diameter"

(12) In the line of 68~70, when using acronyms, professionally, it requires to describe their full name at where the acronyms first appear in a scientific paper.

(13) In the line of 79, the sentence is ambiguous, I guess the "thus guarantee" should be "thus guaranteeing". Otherwise, I don't understand the meaning of the sentence.

(14) In the line of 99, "cell" is a countable noun, either adding a article or changing it into "cells"

(15) In the line of 108~109, change "electrode" into "electrodes"

(16) In the line of 111, please specify "comparing to what" because you are using a comparative sentence.

(17) In the line of 181, cloth is an uncountable noun.

(18) There are also so many other nouns used incorrectly. Please carefully check every noun you used.

(19) In the caption of the Figure 1, "Illustration the synthesis process" should be "Illustration of the synthesis process".

- Format wise

(20) For the Figure 2, the length of scale bars should be specified in the caption, like what you did in figure S6 and S3.

Reviewer #3 (Remarks to the Author):

This manuscript presents a strategic approach for development of a high performance sulfur cathode with an interpenetrated MOF/CNT matrix. The interpenetrated and interweaved CNTs provide a necessary conductivity and structural integrity of the electrode. The highly porous MOFs help to

confine the active sulfur materials and enhance the cyclability of the electrode.

The research idea and approaches are interesting. The results obtained are also reasonable. Therefore, it is recommended to publish this manuscript in Nat. Comm. However, it seems that the authors did not catch up the research pace of the Li-S batteries very well. The high sulfur loading and long cycle life claimed in the manuscript is lower than the state-of-the-art levels that reported within 2015 and 2016. It is suggested the authors to read more recent work from others and claim this study in a proper position. It seems also unfair to compare only with those results with the MOF-based sulfur cathodes. Other flexible sulfur electrodes also showed good cycling performances and high capacity and energy density. Some specific comments are listed below.

1. In Figure 3a, the title of right axis should be "Coulombic efficiency".
2. It is necessary to provide the Coulombic efficiency data along with the discharge capacities, such as in Figure 3 b, c, Figure 4 b, c, d. The Coulombic efficiency data can either be incorporated into the above figures or provided separately.
3. It is necessary to give a note that the volumetric capacity stated in the manuscript was based on the volume of the sulfur cathode rather than the Li-S battery.

REVIEWERS' COMMENTS:

Reviewer #1 (Remarks to the Author):

The revision is fine with me.

Reviewer #2 (Remarks to the Author):

I think that the authored have addressed comments and suggestions very well. It is publishable now.

Reviewer #3 (Remarks to the Author):

The authors answered questions. The manuscript can be published.

Response to reviewers' comments

To Reviewer #1:

This manuscript communicates the making of binder free cathode made from MOFs interpenetrated by CNT. CNT provides the conductivity and also acts as binder, meanwhile they also act as network blocking the lithium sulfides from leaving the MOF cages. The electrode is made flexible with 90 folding causing no significant detrimental effects. The results are quite impressive in terms of capacity and also rate performance for HKUST-1/CNT system. The paper can be accepted for publication with the following issues addressed:

Reply: We are appreciating you very much for your kind recommendation and helpful comments.

1) The pore size differences in MOFs were found to affect the loading of S₈ with larger pores giving better loading as expected. The large pores would have higher chance of lithium sulfides to leak into the electrolyte. Do the authors find this?

Reply: We agree with the reviewer that “larger pore size is generally easier for the loading of S₈ and may make polysulfides easily leak into the electrolyte.” In our cases, however, both pore size and open metal sites of MOFs are responsible for confinement of polysulfides and subsequent cycling performance. For example, MOF-5 with the entrance size relatively smaller than that of HKUST-1, but without open metal sites, shows relatively poorer performance than that of HKUST-1, since HKUST-1 has open copper sites. The relative comments and references are now added on Page 7 in the revised manuscript as “In contrast, HKUST-1 and MOF-5 possess relatively larger entrance size than S₈ molecules, providing easy access for them into pores, which is beneficial for uniform sulfur incorporation and electrochemical reaction. Moreover, the success in sulfur and polysulfides confinement improves electrode cyclability by inhibiting active sulfur loss.^{32, 33}”.

2) For the CNT utilized, do authors find if there are any sulfur inside the tubes? If yes, how much of the contribution is from CNT and how much from MOF?

Reply: This is a good point. However, it is very difficult to exactly distinguish the contribution of the sulfur inside carbon nanotubes and those inside MOFs to the electrochemical performance of S@MOFs/CNT hybrids. From our experiment results, it is clear that the sulfur loaded inside pure carbon nanotubes contributes less to the electrochemical performance, since the much inferior performance of S@ZIF-8/CNT, but the amount of CNT in S@HKUST-1/CNT and S@ZIF-8/CNT are the same. In our case, the performance of the S@HKUST-1/CNT electrodes are derived from the synergistic effect of MOFs and CNTs in the novel hybrid structures.

3) The cycling performance is ok but not fabulous. Would there be any way to improve it?

Reply: This is a good point. We achieved Li-S cells with a high capacity of $\sim 7.45 \text{ mAh cm}^{-2}$ when sulfur loading reached as high as 11.33 mg cm^{-2} by fabricating the S@MOF/CNT electrode, which presents the highest volumetric capacity among the recently reported flexible sulfur cathodes (Fig. 4d and Supplementary Table 3). However, the cycling performance is still far from satisfaction for their practical application. We think that the main obstacle to achieve better performance lies in the imperfection in preventing the shuttling effect for this hybrid material due to the relatively large entrance of the selected MOFs, i.e. HKUST-1 and MOF-5. It might be possible to find a MOFs with proper entrance size but big pore volume with plenty open metal sites, which could not only let solvated Li ions transfer in and out, but also allow sulfur species enter the pore and confined in the pores more effectively. The related comment is added in the penultimate paragraph of discussion section as “The cycling performance of sulfur electrodes with high volumetric energy densities can be further improved by finding MOFs with proper entrance pore size, big pore volume, and plenty of open metal sites, which not only let solvated Li ions transfer in and out, but also allow sulfur species enter to and confined in the pore for better inhibition of the polysulfide shuttle”.

To Reviewer #2:

In this paper, the authors disclosed a new strategy to interpenetrate the MOFs with CHTs as a foldable and bendable film for the cathode of Li-S batteries. By virtue of the electrostatic interaction between MOFs and polysulfides, the porous structure of MOFs, and the integrity of interweaved MOF/CNT film, the performances of Li-S batteries could be effectively enhanced. In addition, the film is binder-free, which is beneficial for the energy density of Li-S batteries. The experimental results are interesting and impressive. The novelties include: (i) this study made a comparison of three MOFs with different pore window and claimed HKUST-1 has suitable pore size to store S₈ molecule; and (ii) the Li-S batteries showed the best reported performance in MOF based sulfur electrode. Based on the novelties and current quality of your manuscript, I think that this manuscript can be published on Nature Communications after some suggestions and comments are addressed:

Reply: We are appreciating the reviewer very much for positive recommendation and constructive comments to improve our work.

• Technical questions

(1) Does CNTs interpenetrate through MOFs? From your SEM image (Fig.2d), it looks like the CNT only weaves MOF crystals and doesn't penetrate through the MOFs. If so, kindly provide cross-section TEM images showing the situation. All your discussions are based on the structure, you could better

make it reliable and dependable.

Reply: This is a very good point. As suggested, we did the experimental again and show the SEM images for the cross-section of cracked hybrid thin film to confirm interpenetrated structure of MOFs/CNT in supplementary Fig. S3a and S3b, here Figure R1a, b. We also tried to record the cross-section TEM image, but the result is not clear due to the heavy of Cu based MOFs and low contrast of CNTs. In order to see CNTs interpenetrate through MOFs crystals by TEM clearly, an indirect way was tried. We found that morphology and structures of the MOFs/CNT thin film could be preserved after carbonizing the MOFs/CNT at high temperature (800 °C) in N₂ gas for 2-3 hours and removing away the resulted metal or metal oxide species by diluted nitric acid. The corresponding TEM result of the carbonized hybrid HKUST-1/CNT thin film after removing away copper or copper oxide species is shown in (Supplementary Figure 3c, here Figure R1c), which confirms that the CNTs interpenetrate through MOFs. All the above results strongly support the penetration of CNT through MOFs in our work.

(2) As described, elementary sulfur exists in the cavity of MOFs, especially in the cases of HKUST-1 and MOF-5, and CNTs weaves the MOF crystal (500 nm ~ 2 μm), how come the S₈ inside the MOFs gets the electrons and Li⁺ ions to go through electrochemical reactions? Can the introduction of CNTs dramatically improve the conductivity of MOF matrix? In my opinion, MOFs are extremely insulative (10⁻¹⁰ S/cm), the distance of electron tunneling in MOFs is also very short (at the nanometer long). So the only reactive sites at the interface between CNTs and MOF crystals could be useful for the electrochemical reactions.

Reply: This is a very good point. It is no doubt that both the electronic conductivity and ionic transfer are responsible for cycling performance of the S@MOFs/CNT electrodes. Although MOFs are very insulative (10⁻¹⁰ S/cm), the introduction of CNTs significantly improves the conductivity of MOFs matrix. The penetration of CNTs in HKUST-1 crystals offers very small resistance (i.e., 35 Ω/□ by four-point measurement), which is also confirmed by the greatly improved impedance of S@HKUST-1/CNT electrode as shown in Supplementary Fig. 6b. Moreover, due to the transport of solvated Li ions in and out of the pores of MOFs, they supply enough ionic conductivity for the occurrence of sulfur electrochemistry inside the MOFs. As a result, the electrochemical reactions of sulfur species mainly happen in reactive sites at all the interface between CNTs and MOF crystals, where electronic and ionic conductivities are guaranteed.

(3) What is the amount of CNTs in the sample of S@HKUST-1/CNT with 70% sulfur loading for batteries? Both sulfur and MOF are highly non-conductive, without enough CNTs as conductive networks, please explain how come the electrode can deliver a high capacity over 500 mAh/g (Table S4)?

Reply: Based on the calculation, the content of CNTs in the S@HKUST-1/CNT with 70% sulfur loading is around 12% (We add this information in the experimental section on Page 12 in the revised manuscript). It is no doubt that the content of CNT is critical for the cycling performance of the S@MOFs/CNT electrodes shown in Supplementary Fig. 6. Due to the relatively low content of in the electrode with 70% sulfur loading, its capacity is much lower than that of the electrode with 40% sulfur loading. However, it needs to be noted out that the S@HKUST-1/CNT electrodes with 70% sulfur were cycled within an enlarged potential window of 1.5-3.0 V (vs. 1.8-2.6 V for electrodes with 40% sulfur loading) to overcome large overpotential for more capacity (We also add this information in the experimental section on Page 12 in the revised manuscript). As a result, the S@HKUST-1/CNT electrode with 70% sulfur loading still delivers a capacity over 500 mAh g⁻¹ due to its unique CNT interpenetrated MOFs and well interconnected CNTs networks with hierarchical porous structures within an enlarged voltage window.

(4) For element mapping images shown in Figure S3, the corresponding EDX analyses should be accompanied as well, because the elemental relative content is very sensitive to the time to collect the characteristic X-ray signal. Technically and precisely, only elemental mapping images cannot demonstrate that sulfur is less in S@ZIF-8/CNTs after CS₂ washing. Besides, a question for the method of CS₂ washing is raised up, the sulfur incorporation into the cavities of MOFs was achieved by CS₂ solution. To remove the sulfur dispersed on the surface of S@ZIF-8/CNTs, the same method is also used. How can you ensure that the sulfur inside the cavities of composites was not washed? I think the sulfur can go in and also can go out providing using the same method.

Reply: Thanks for your kind suggestion. We add the corresponding EDX analyses spectra and weight percents of the corresponding elements of Figure S4a-f in the revised manuscript as shown in Supplementary Fig. 4g and Supplementary Table 4, which are in consistent with the element mapping results. “The variation of peak intensities (Supplementary Fig. 4g) and element contents (Supplementary Table 4) before and after CS₂ washing also strongly support this result.” The related discussion is now added on Page 4 in the revised manuscript.

We agree with the reviewer that “during the washing, the sulfur S₈ can also go out from the MOFs.” However, the preparation process for the S@MOFs/CNT electrodes is different from the washing process. A long-time thermal treatment for preparation of the S@MOFs/CNT electrodes was subsequently performed after the S/CS₂ solution infiltration process for improved sulfur incorporation. This process was described in detail in the experiment section of “Preparation of S@MOFs/CNTs thin films” as “Typically, sulfur CS₂ solution (5 mg ml⁻¹) was first prepared by dissolving sublimed sulfur in CS₂ solvent under stirring at room temperature. The obtained solution was dropped into the MOFs/CNT substrates and slowly dried at 40 °C. The sulfur contained film was further dried under

vacuum at 60 °C for 8 hrs and subsequently annealed at 140 °C for 8 hrs under Ar atmosphere and cooled to room temperature to obtain the S@HKUST-1/CNT, S@MOF-5/CNT, and S@ZIF-8/CNT electrodes, respectively.” As comparison, only several times of quickly CS₂ washing could easily clean the superficial sulfur, which keep the sulfur inside the pores as enough as possible. In addition, since the loading amount of sulfur, the loading process, and the washing process were the same, the less amount of sulfur remained in the S@ZIF-8/CNT and S@MOF-5 electrodes than those in the S@HKUST-1/CNT electrode indicates the proper entrance size and pore structures of HKUST-1 is more suitable for sulfur confinement. This is also confirmed by their corresponding electrochemical performances.

(5) In Figure S5(c), there is a discrepancy. The cycling stability of S@HKUST-1/CNT (3:0.4) is worse than that of S@HKUST-1/CNT (3:1.2). However, the relative amount of HKUST-1 of the former is higher than that of the latter, meaning S@HKUST-1/CNT (3:0.4) should have better capabilities in confining polysulfides. As a consequence, S@HKUST-1/CNT (3:0.4) should show the better cycling stability than that of S@HKUST-1/CNT (3:1.2), which is discrepant from the results shown in Figure S5(c). Please double check.

Reply: Thanks for your comments. We double check the results and make sure that Supplementary Fig. 6c (previous Supplementary Fig. 5c) is correct. Typically, cycling stability is influenced by not only the confinement of sulfur species but also the conductivity of electrodes. A balanced ratio of HKUST-1 to CNT contributes to the best cycling stability of the S@HKUST-1/CNT electrode, i.e., the S@HKUST-1/CNT (3:2) electrode. Since the S@HKUST-1/CNT (3:1.2) electrode possesses a closer value to the best ratio of 3:2, it is reasonable that it exhibited a better cycling stability than that of the S@HKUST-1/CNT (3:0.4) electrode.

(6) You defined 1C = 1675 mAh/g, the unit should be “mA/g” not “mAh/g”. please correct it.

Reply: Thanks. The unit is corrected into “mA g⁻¹” in the revised manuscript.

• Grammatical and spelling mistakes

(7) In the line of 14 in the abstract, “candidate” should be “candidates” or “a promising candidate” because this is a countable noun.

Reply: Thanks. We are sorry about our carelessness. These grammatical and spelling mistakes are corrected as yellow marked in the revised manuscript, which makes it a much better revision.

The “candidate” is corrected into “candidates” in the revised manuscript.

(8) In the line of 24, “A, B and C” should be “A, B, and C”. In the line of 27, “A, and B” should be “A and B”, It don't need a comma between A and B.

Reply: Thanks. We double check the whole manuscript and correct such errors in the revised manuscript.

(9) In the line of 27, “Flexible energy storage system” should be “A flexible energy storage system” or “Flexible energy storage systems”. Because “system” is a countable noun

Reply: As suggested, this sentence is corrected into “Flexible energy storage systems are...” in the revised manuscript.

(10) In the line of 46, “flexible even foldable” should change into “flexible and even foldable”

Reply: As suggested, the “flexible even foldable” is corrected into “flexible and even foldable” in the revised manuscript.

(11) In the line of 62, there is a typo error, “tinny diameter” should be “tiny diameter”

Reply: As suggested, the “tinny diameter” is corrected into “tiny diameter” in the revised manuscript.

(12) In the line of 68~70, when using acronyms, professionally, it requires to describe their full name at where the acronyms first appear in a scientific paper.

Reply: As suggested, the full names of the acronyms are added in the revised manuscript.

(13) In the line of 79, the sentence is ambiguous, I guess the “thus guarantee” should be “thus guaranteeing”. Otherwise, I don't understand the meaning of the sentence.

Reply: As suggested, the “thus guarantee” is corrected into “thus guaranteeing” in the revised manuscript.

(14) In the line of 99, “cell” is a countable noun, either adding a article or changing it into “cells”

Reply: As suggested, the “cell” is correct into “cells” as yellow marked in the revised manuscript.

(15) In the line of 108~109, change “electrode” into “electrodes”

Reply: The “electrode” here refers to the specific electrode i.e. S@CNT or S@HKUST-1+CNT, we add “the” in front of them to make it clear in the revised manuscript.

(16) In the line of 111, please specify “comparing to what” because you are using a comparative sentence.

Reply: As suggested, this sentence is revised into “...the S@HKUST-1 electrode exhibits more stable cycling behavior but lower capacity than S@HKUST-1+CNT electrode due to the...” in the revised manuscript.

(17) In the line of 181, cloth is an uncountable noun.

Reply: This sentence is revised into “...weave the electrode into pliable and tough cloth to....” in the revised manuscript.

(18) There are also so many other nouns used incorrectly. Please carefully check every noun you used.

Reply: Thank you very much. As suggested, we carefully check the whole manuscript and make sure every noun used correctly.

(19) In the caption of the Figure 1, “Illustration the synthesis process” should be “Illustration of the synthesis process”.

Reply: Thanks again. The caption of Figure 1 is revised into “Illustration of the synthesis process” in the revised manuscript.

• Format wise

(20) For the Figure 2, the length of scale bars should be specified in the caption, like what you did in figure S6 and S3.

Reply: Thank you very much for this kind reminding. We specify the scale bars in the caption of all the Figures in the revised manuscript.

To Reviewer #3:

This manuscript presents a strategic approach for development of a high performance sulfur cathode with an interpenetrated MOF/CNT matrix. The interpenetrated and interweaved CNTs provide a necessary conductivity and structural integrity of the electrode. The highly porous MOFs help to confine the active sulfur materials and enhance the cyclability of the electrode.

The research idea and approaches are interesting. The results obtained are also reasonable. Therefore, it is recommended to publish this manuscript in Nat. Comm. However, it seems that the authors did not catch up the research pace of the Li-S batteries very well. The high sulfur loading and long cycle life claimed in the manuscript is lower than the state-of-the-art levels that reported within 2015 and 2016. It is suggested the authors to read more recent work from others and claim this study in a proper position. It seems also unfair to compare only with those results with the MOF-based sulfur cathodes. Other flexible sulfur electrodes also showed good cycling performances and high capacity and energy density. Some specific comments are listed below.

Reply: We appreciate the reviewer's positive evaluation on our work. We also thank the reviewer much for kind recommendation and nice comments. We compared our results with those of MOF-based sulfur cathodes (Supplementary Table 1), but also representative high-performance S@carbon electrodes (Supplementary Table 2), and flexible sulfur electrodes (Supplementary Table 3) in the previous manuscript. As suggested, we read more recent work published in 2015 and 2016 and update related references in Supplementary Tables 1, 2 and 3 (highlighted by yellow) for more comprehensive performance comparisons in the revised manuscript. Based on the new comparisons, we can conclude that: 1. Long-term cycling life of our S@HKUST-1/CNT electrodes with high sulfur loading are still among the top level of reported sulfur electrodes (Supplementary Tables 1 and 2). 2. The areal capacity of our electrode is also among the top level of the reported sulfur electrodes (Supplementary Table 3). 3. The volumetric energy density of our electrodes is higher than those reported of flexible sulfur electrodes (Supplementary Table 3). All the above results show great potential applications of our MOFs-based materials for flexible Li-S batteries.

1. In Figure 3a, the title of right axis should be "Coulombic efficiency".

Reply: Thanks. The right axis of Figure 3a has been changed to "Coulombic Efficiency" in the revised manuscript

2. It is necessary to provide the Coulombic efficiency data along with the discharge capacities, such as in Figure 3 b, c, Figure 4 b, c, d. The Coulombic efficiency data can either be incorporated into the above figures or provided separately.

Reply: Thanks for your kind suggestion, the Coulombic efficiency data have been added in Figure 3b, c and Figure 4b, c, d in the revised version.

3. It is necessary to give a note that the volumetric capacity stated in the manuscript was based on the volume of the sulfur cathode rather than the Li-S battery.

Reply: As suggested, we add the statement of “Volumetric capacities were calculated based on the volume of sulfur electrodes” in the experimental section on Page 13 in the revised manuscript.

[Answers to Editorial Requests]

EDITORIAL REQUESTS:

* Nature Communications uses a transparent peer review system, where for manuscripts submitted from January 2016 we are publishing the reviewer comments to the authors and author rebuttal letters of our research articles online as a supplementary peer review file. Please let us know in the cover letter when submitting the final version of your manuscript whether you are opting out of this scheme or not. If you are concerned about the release of confidential data, we do permit redactions in the interest of confidentiality. If you would like to remove such information from these files, then please let us know specifically what information you like to have removed. Please note that we cannot incorporate redactions for other reasons.

Answer: We are agree to publish the reviewer comments to the authors and author rebuttal letters of our research articles online as a supplementary peer review file.

* Your manuscript should comply with our policies and format requirements, detailed in our checklist for authors at:http://www.nature.com/article-assets/npg/ncomms/authors/ncomms_manuscript_checklist.pdf

Answer: We have carefully checked and the final version of our manuscript complies with your policies and format requirements. The corresponding checklist is uploaded with this submission.

* Please also review the changes in the attached copy of your manuscript, which has been edited for style, and address the comments and queries I have added. If using Word, please use the 'track changes' feature to make the process of accepting your manuscript more efficient.

Answer: We have carefully addressed all the issues which you raised and marked them in the final version. The reference 59 is the same as reference 31, which has been removed. The sequences of the corresponding references have been reordered.

* Data availability statements and data citations policy: All Nature Communications manuscripts must include a section titled "Data Availability" at the end of the Methods section or main text (if no Methods). For more information on this policy, and a list of examples, please see

<http://www.nature.com/authors/policies/data/data-availability-statements-data-citations.pdf>

- Accession codes for deposited data
- Other unique identifiers (such as DOIs and hyperlinks for any other datasets)
- At a minimum, a statement confirming that all relevant data are available from the authors

-
- If applicable, a statement regarding data available with restrictions
 - If a dataset has a Digital Object Identifier (DOI) as its unique identifier, we strongly encourage including this in the Reference list and citing the dataset in the Data Availability Statement.

Answer: The Data Availability statement has been added as

“Data Availability:

All relevant data are available from the authors.” At the end of the Method section.

* Please check whether your manuscript contains third-party images, such as figures from the literature, stock photos, clip art or commercial satellite and map data. We strongly discourage the use or adaptation of previously published images, but if this is unavoidable, please request the necessary rights documentation to re-use such material from the relevant copyright holders and return this to us when you submit your revised manuscript.

We are committed to ensuring clarity and avoiding ambiguity in the mathematics in our papers. Consequently, please carefully check the mathematical terms throughout your manuscript (including labels on figures and figure captions) to ensure that it conforms strictly to the following guidelines. In mathematical terms, scalar variables (e.g. x , V , γ) and constants (e.g. π , h , e) should be typeset in italics, and vectors (such as r , the wavevector k , or the magnetic field vector B) should be typeset in bold without italics. In contrast, subscripts and superscripts should only be italicized if they too are variables or constants. Those that are labels (such as the 'c' in the critical temperature, T_c , the 'F' in the Fermi energy, E_F , or the 'crit' in the critical current, I_{crit}) should be typeset in roman. Please also ensure the same convention is followed in figure labels, axes, and such. Additionally, to avoid doubt, unit dimensions should be expressed using negative integers (e.g. $\text{kg m}^{-1} \text{s}^{-2}$ not kg/ms^2) or the word 'per'.

If a consortium is included in the main author list, all members of the consortium are considered bona fide authors, and must be listed together with their affiliations at the end of the main Article (not in the Supplementary Information). However, if a member of the consortium already appears as an individual name in the main author list, then his/her name should not be listed again at the end of the Article. If you need to give credit to a consortium, a project or a group of people who do not meet authorship criteria, you can add a mention in the Acknowledgements section or elsewhere (in which case, a full list of members can be provided as a Supplementary Note in the Supplementary Information, if desired). For guidelines on authorship and consortia, please visit: <http://www.nature.com/authors/policies/authorship.html>

Answer: Our manuscript does not contain third-party images. The mathematical terms throughout our manuscript (including labels on figures and figure captions) are strictly to the Nature Communications formats. No consortium

is included in the author list.

* Your paper will be accompanied by a two-sentence editor's summary, of between 250-300 characters, when it is published on our homepage. Could you please approve the draft summary below or provide us with a suitably edited version.

Lithium sulfur batteries show capacity and cost advantages, but suffer from the insulating sulfur, shuttling effect and volume fluctuation. To address these challenges, the authors synthesize foldable composite electrodes with carbon nanotubes interpenetrating metal-organic frameworks.

Answer: We think this summary is pertinent and perfect!

Above all are the editorial requests and corresponding answers. Thank you very much!